# WIQOR: A DATASET FOR WHAT-IF ANALYSIS OF OPERATIONS RESEARCH PROBLEMS

## ABSTRACT

We formalize the *mathematical program modification* (MPM) task, in which the goal is to revise a mathematical program according to an inquiry expressed in natural language. These inquiries, which we refer to as *what-if questions*, express a desire to understand how the optimal solution to an optimization problem changes with the addition, deletion or revision of constraints. In detail, each MPM instance is a triple consisting of: 1) a natural language *specification* that summarizes an optimization problem, 2) the *canonical* formulation of the problem, and 3) a natural language what-if question. The goal is to predict the updated canonical formulation with respect to the question. To support the study of this task, we construct WIQOR, a dataset of 1,946 MPM instances, derived from NL4OPT (Ramamonjison et al., 2023), but with the number of decision variables extended to more than 30 for some problems. In experiments, we observe that Llama 3.1 70B instruct under the in-context learning paradigm achieves 69% accuracy on the easiest test instances, but only 36% accuracy on the most complicated problems. We release WIQOR in the hopes of spurring additional study of MPM and ultimately enabling non-technical users to conduct what-if analyses without the help of technical experts.

## 1 INTRODUCTION

Mathematical programming is the centerpiece of decision making in a many industries. For example, firms routinely employ mathematical optimization to set product prices (Ferreira et al., 2016), find optimal transportation routes (Holland et al., 2017; Dang et al., 2024), or to optimize their supply chain operations – from optimal loading of a single truck to the best location for a new warehouse (Mehrotra et al., 2024). Tools for mathematical optimization are commonly utilized in academic peer review in order to match papers to reviewers (Taylor, 2008; Stelmakh et al., 2019). Real-world optimization problems are massive and complex; they can have more than hundreds—or even thousands—of variables and constraints. For example, UPS optimizes routes of 55,000 drivers across the United States (Holland et al., 2017).

Due in part to the complexity of real-world optimization problems, constructing formal problem representations that can be solved by industry-grade optimizers is typically a cumbersome process requiring collaboration between two parties: a domain expert and a technical expert (Li et al., 2023; Mostajabdaveh et al., 2024). In these cases, the domain expert, e.g., a vendor who seeks to optimize their product prices, does not possess the requisite technical expertise to translate their specific problem into a mathematically rigorous objective function, decision variables, and the associated set of constraints. Instead, the domain expert provides a detailed account of their problem to a technical expert, who uses these details to construct a problem representation that can be sent as input to an off-the-shelf solver (e.g., Gurobi), which returns the solution (Gurobi Optimization, LLC, 2024).

In addition to posing a challenge during program construction, the necessary communication between the domain and technical experts slows analysis of the program and its optimized solution. In many cases, to maintain a good understanding of the optimal solution it is common to perform counterfactual analysis, which usually consists of modifying the program slightly, re-running modified problem through the solver, and comparing the original and new solution (Li et al., 2023). For example, consider that upon inspection of set of optimized prices, the domain expert seeks to understand how a new constraint on the maximum price for a given product would affect the optimal

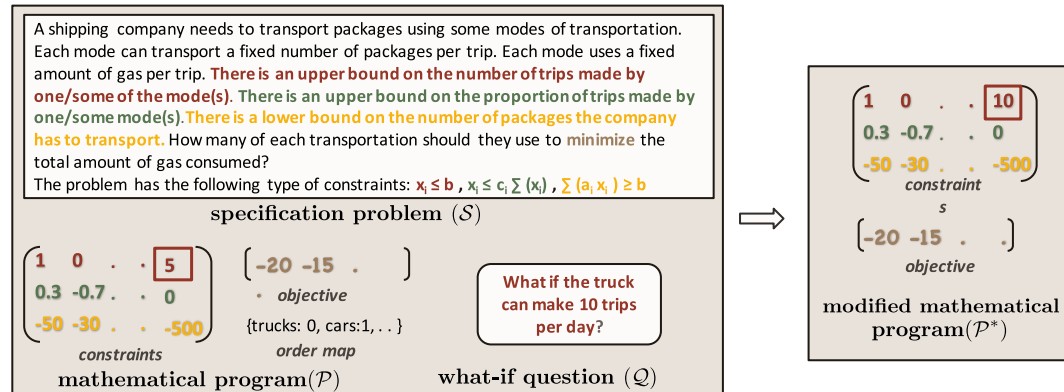

Figure 1: A sample MPM instance from the WIQOR dataset. The left box shows the input triple consisting of problem specification and it's mathematical program and a what-if question, while the right side demonstrates the modified program based on that what-if question. Two of the transportation modes used for this example are truck and car, with the ellipsis signifying potential for existence of additional variables. Here the what-if question changes the limit on the upper bound of truck trips from 5 to 10, as seen in the modified first row of the mathematical program's canonical form.

prices. In such cases, the domain expert must transmit the new constraint to the technical expert, who then modifies the initial program and solves it anew, transmitting the optimal solution back to the domain expert. Needless to say, reliance on the technical expert significantly slows analysis and may lead the domain expert to skip the analysis entirely.

In order to support the development of methods that enable non-technical users to engage in such analysis, we formalize and study a task we term *mathematical program modification*, MPM. In this task, the goal is to update an existing mathematical program according to a natural language inquiry. In detail, the input in MPM is a triple consisting of: i) a natural language *specification*, or summary, of an optimization problem, ii) the problem expressed formally (mathematically) in *canonical form*, and iii) a natural language inquiry about how a desired change in the constraints would modify optimal solution to the problem, which we call a *what-if question*. The goal of the task is to correctly modify the provided canonical form with respect to the what-if question.

To facilitate the study of this task, we present the WIQOR (**W**hat-**I**f **Q**uestions for **O**perations **R**esearch problems) benchmark dataset. This dataset comprises 1,946 instances of the MPM task, and includes what-if questions that correspond to 4 different types of constraint changes (Table 1), which can be applied to 7 different constraint types (Appendix table 4). Each MPM instance in WIQOR is seeded by a canonical formulation that is borrowed from the NL4OPT dataset Rama-monjison et al. (2023). The accompanying problem specification, what-if question, and updated canonical formulation are constructed using a combination of heuristics and large language model inference. All MPM instances undergo manual evaluation before being added to the dataset.

To better capture real-world complexity, WIQOR includes 4 test splits with increasing complexity— as measured by the number of decision variables and constraints. In the `Test-Base` (easiest) test split, MPM problems have on average 2.08 decision variables and 2.63 constraints. However, WIQOR also includes additional test split `Test-VarAug` which include 400 base problems that have been extended by adding 5, 10, 20 or 30 (100 problems each) additional decision variables and corresponding constraints, respectively. Expansion of canonical form to include additional variables and constraints is accomplished using a sequence of carefully designed heuristics and logic.

Empirically, we evaluate the Llama 3.1 instruct family of models on the WIQOR dataset under the in-context learning paradigm. Our results reveal that the smallest models struggle to achieve more than 40% accuracy on the base test instances. As the number of decision variables increase, even the 70 billion parameter model shows a decline in performance, achieving only 36% accuracy on the most difficult split. We hope that WIQOR helps support the development of tools that will empower domain experts conduct what-if analysis without the need to enlist the help of external technical experts.

## 2 RELATED WORK

**LLMs for Operations Research** A foundational work at the intersection of operations research and NLP is the NL4Opt dataset by Ramamonjison et al. (2023), which focuses on the task of translating optimization problems into mathematical formulations. This dataset has problems from various domains, and the problems are written in natural language with a description of the constraints and the target optimization. The inputs for the NL4Opt subtask-2, i.e. the generation task, are the problem description, its set of problem entities, and the order mapping of variables in the columns of the mathematical program written in canonical form. The ground-truth label annotations consist of the objective declaration and the constraints declarations, which are converted to a meaning representation using a semantic parser, which are in turn converted to a canonical form for evaluation. While NL4Opt is an important step forward, it falls short in reflecting the types of optimization problems that real-world operations research experts encounter in industry and academia. Everyday optimization tasks are often more complex and varied, and the challenges experts face are not fully captured by this dataset.

Additional efforts have been made to develop a variety of datasets aimed at advancing research in this field. AhmadiTeshnizi et al. (2024) release NL4LP, a dataset of long and complex OR problems. They propose a scalable system where LLMs develop mathematical models for mixed integer linear programming (MILP) problems. Xiao et al. (2024) released ComplexOR a dataset of complex operations research problems and explore how LLM agents can be combined in expert chains to tackle these problems. However, these newly developed datasets are quite limited in size, making them unsuitable for training models on the tasks they define.

There have also been efforts to synthesize operations research datasets using language models. Prasath & Karande (2023) evaluate the use of CodeT5 for synthesizing mathematical programs from natural language, employing data augmentation, beam post-processing, and back translation via GPT-3 to generate synthetic examples. Yang et al. (2024) introduce another benchmark: E-OPT, designed to evaluate LLMs' ability to solve complex optimization problems, extending beyond simple linear programming to include nonlinear problems. They propose the Reverse Socratic Synthesis (ReSocratic) method, which incrementally synthesizes mathematical formulations and back-translates them into problem descriptions. This strategy of reverse generating problem scenarios is akin to how what-if questions in the WIQOR dataset were generated (subsection 4.3).

Mostajabdaveh et al. (2024) also propose a framework to model real-world optimization problems from natural language specifications. Similar to our work, they recognize that real-world problem descriptions rarely contain declarations of all the parameters and variables with keywords identifying them. While their goal is to formulate mathematical problems based on the abstracted problem specification only (which does not contain any numerical values), our task supplies the numerical values *in addition* to the problem specification, in a succinct (matrix) format, and the challenge of our task is in translating a natural language modification inquiry to an appropriate modification of that matrix.

While these previous works have primarily focused on the task of generating mathematical programs from various formats of problem descriptions, none have addressed the challenge of modifying an existing mathematical program in response to changes expressed in natural language, such as a *what-if* question. This paper seeks to address this gap by formalizing the Mathematical Program Modification MPM task and introducing WIQOR, a benchmark dataset specifically designed to evaluate LLMs' ability to modify mathematical programs based on natural language modification inquiries. Additionally, we provide baseline performance results for various LLMs, aiming to spur further research in this area.

**Counterfactual reasoning with LLMs** Large Language Models (LLMs) have shown increasing proficiency in counterfactual reasoning, which involves understanding and generating hypothetical scenarios. Chen et al. (2023) focus on distilling LLMs' ability to reason about counterfactual scenarios, proposing methods to improve the model's understanding and generation of counterfactual statements. Similarly, Qin et al. (2019) explore their capacity to generate coherent and plausible alternative realities within narrative contexts, demonstrating that the model can construct counterfactual versions of stories that remain logically consistent with the original premises. Additionally, Qin et al. (2023) provide a formal evaluation of LLMs' performance in counterfactual tasks, revealing limitations in the model's ability to fully grasp the nuances of hypothetical scenarios. These

studies collectively highlight both the promise and challenges of LLMs in handling counterfactual reasoning, making this line of research highly relevant to the task of modifying mathematical programs based on natural language modification inquiries in counterfactual contexts, as discussed in our work.

# 3 THE MATHEMATICAL PROGRAM MODIFICATION TASK (MPM)

Recall that our goal is to facilitate what-if analysis (of mathematical programs) by non-technical users. That is, we aim to enable domain experts to analyze how the optimal solution to their optimization problems changes subject to various modifications; all modifications supplied via natural language. As a first step, we formulate the *mathematical program modification* (MPM) task. In an instance of MPM, the input is a triple consisting of: i) a *specification*, $\mathcal{S}$, i.e., brief summary of an optimization problem, including classes of variables and constraints; ii) the corresponding problem formalized mathematically, $\mathcal{P}$; and iii) a *what-if question*, $\mathcal{Q}$, expressed in natural language. The goal in MPM is to modify $\mathcal{P}$ according to $\mathcal{Q}$. Formally,

$$\text{MPM} := (\mathcal{S} \times \mathcal{P} \times \mathcal{Q}) \to \mathcal{P}^\star \tag{1}$$

where $\mathcal{P}^\star$ is the modified mathematical program. In the following subsections, we detail each component of an instance of MPM, define notation, and introduce associated terminology.

## 3.1 OPTIMIZATION PROBLEM SPECIFICATIONS

Real-world optimization problems include hundreds of decision variables and constraints, or more. When communicating about such problems, it is common to construct a *specification*, or summary of the problem. Rather than detail each variable and constraint, specification often mentions the groups of variables that appear in a problem as well as describe the relationships between variables and constraints. An example of a problem specification appears in Figure 1. Notice that the specification for the problem includes the fact there is an upper bound on the number of each type of shipping that can be utilized, but it does not detail the precise upper limit for each shipping type. Please refer to figure 3 for an example of how various details from an optimization problem are converted into a specification problem.

## 3.2 CANONICAL FORMULATION

The problem specification summarizes an underlying (constrained) optimization problem. However, the specification does not include enough information to solve the problem. Specifically, each program we study includes an *objective function*, a collection of *decision variables* and a set of *constraints* (4)—none of which are fully detailed in the problem specification.

Instead, all details necessary to solve a given optimization problem are included in that problem's *canonical formulation*, $\mathcal{P}$. The canonical formulation is comprised of 3 objects: i) the objective function, ii) the constraint matrix, and iii) an *order mapping*. In a canonical formulation, the objective is always formulated as a maximization of a linear combination of decision variables. Additionally, the constraint matrix is written as $Ax \leq b$, where $A_{ij}$ is the coefficient corresponding to decision variable $j$ in constraint $i$, $x$ is a vector of decision variables, and $b$ is a vector of constants. To rewrite any objective or constraint in canonical form, we perform algebraic manipulations as necessary. Since the MPM task requires translating natural language inquiries about specific variables into modifications of the constraint matrix, we require an order mapping, which maps variable names to their indices in $x$ (or, equivalently, the columns of $A$). Without this map, applying the modifications suggested by a what-if question would be impossible. We note that the output of the MPM task, $\mathcal{P}^\star$, is also a canonical formulation of an optimization problem. In particular, $\mathcal{P}^\star$ is identical to $\mathcal{P}$ except for the modifications implied by the what-if question, $\mathcal{Q}$.

## 3.3 WHAT-IF QUESTIONS

Ultimately, our goal is to enable domain experts to perform certain analyses of mathematical programs via natural language. Toward this end, we define a *what-if question*, $\mathcal{Q}$, to be natural language inquiry regarding how the optimal solution to a mathematical program changes if some of its constraints change. As the name suggests, all such inquiries begin with the hypothetical marker phrase

*What if*, as in: *What if the price of product $p$ must be greater than $x$?* Any what-if question can be answered by modifying the constraint matrix in the way implied by the question, and resolving the problem. We say that each what-if question has a *type*, which is determined by the type of change it imples. In our work, we study what-if questions of 4 types:

1. **Limit change (LC):** Questions of this type require an alteration of the bounds of a constraint. For example, *What if the maximum number of truck trips in a week was 10 instead of 5?*

2. **Constant change (CC):** Questions of this type require a modification of a constant parameter value within the constraint. For example, *What if the car can transport 50 packages per trip?*

3. **Constraint direction reversal (CDR):** Questions of this type require a modification of the direction of a constraint, such as changing an upper bound to a lower bound constraint. For example, *What if the truck should make at least 5 trips in a week instead of at most 5 trips?*

4. **Variable interaction modification (VIM):** Questions of this type require a modification of how variables interact within a constraint, altering the sign of a variable's coefficient. For example, *What if, instead of the total number of packages transported by the truck and car being at least 500, the difference between the number of packages transported by the truck and the car had to be at least 500?* (e.g., changing $50t + 30c \geq 500$ to $50t - 30c \geq 500$).

## 4 THE WIQOR DATASET

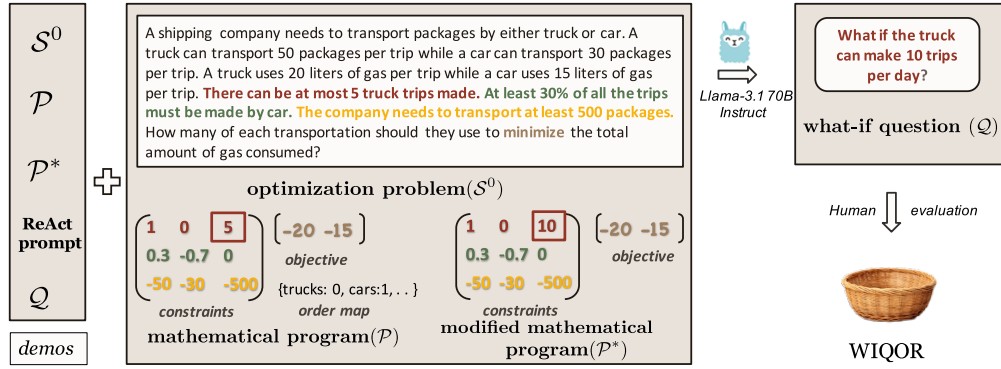

Figure 2: **Reverse Engineering What-if Questions.** What-if questions are generated by a large language model using the in-context learning paradigm. The model is provided with 3 demonstrations of the task which include: i) the original optimization problem from NL4OPT, $S^0$, ii) the canonical formulation of the problem, $\mathcal{P}$, iii) the modified canonical formulation, $\mathcal{P}^\star$, iv) a ReAct reasoning string, and v) the corresponding what-if question, $\mathcal{Q}$. Following these demonstrations, we supply the test problem, which matches the demonstrations except that it is missing the what-if question. The model is thus prompted to generate a what-if question, which is manually evaluated before inclusion in WIQOR.

In this section, we detail the construction of the WIQOR, the first dataset for the MPM task. Starting from a seed problem in NL4OPT Ramamonjison et al. (2023), we utilize a combination of heuristics, large language model (LLM) inference, and human evaluation to construct and filter all components of each MPM instance in WIQOR. Finally, we discuss dataset statistics, focusing on problem types and complexity, as measured by the number of decision variables.

### 4.1 OPTIMIZATION PROBLEMS IN CANONICAL FORM

At the heart of each MPM instance in WIQOR is an optimization problem, written in canonical form. To construct these canonical formulations for WIQOR, we use—and optionally modify—problems from NL4OPT. NL4OPT is a dataset of optimization problems, where the goal is to take a natural language description of the optimization problem and generate the corresponding objective

function and constraint matrix, in canonical form (Section 3.2). For each problem in NL4OPT, we utilize the target canonical formulation and provided order map.

**Increasing complexity.** Real-world optimization problems may include hundreds or thousands of decision variables and constraints. In contrast, NL4OPT has an average of 2.08 variables and 2.63 constraints per problem. Therefore, we augment MPM instances in the test split of the WIQOR with 5, 10, 20, or 30 new decision variables. At a high level, this is done by adding columns to the constraint matrix, and making corresponding changes to the order map. When adding new variables, we also modify existing constraints and add new constraints that resemble those that are already present. For example, if each variable in the original problem has an upper bound constraint, we introduce new upper bound constraints for the added variables. As another example, we expand existing sum constraints with new variables as well. To update the order map with the new variables, we assign names to the new variables based on the original variable names. For instance, if the original variables were `truck` and `car`, and we are adding three new variables, we randomly select one of the original names (either `truck` or `car`) and append the suffix `-idx` to it, where `idx` is an integer in the range from 0 to $k - 1$ (in this example, $k = 3$). This process generates variable names like `truck-0`, `car-1`, `truck-2`, and so on. Significant care must be taken so that the resulting problems are sensible. For a detailed accounting of our handling of corner cases, refer to the algorithm in Appendix A.5.

## 4.2 PROBLEM SPECIFICATIONS

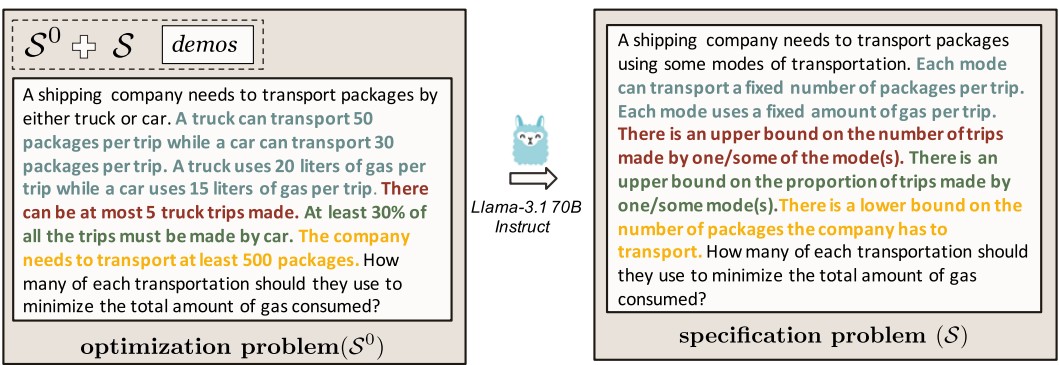

Figure 3: Problem specifications include the type of constraints and target optimization, omitting variable names and their counts as well as parameter values associated with the constraints.

In addition to optimization problems in canonical form, each MPM instance in WIQOR includes a problem specification. Recall that these specifications are natural language summaries of the problems (Section 3.1). Unlike the NL4OPT problems, our specifications do not explicitly name *any* of the decision variables or detail the constraints. In other words, there is insufficient information in a specification to generate the canonical form of the problem. Importantly, by virtue of being summaries, these specifications need not be modified when canonical formulations are extended with new variables.

We generate specification problems using a large language model (LLM). To do so, we begin by manually creating 4 specifications to accompany the original problems in NL4OPT. We combine an optimization problem expressed in natural language from NL4OPT with the corresponding hand-written specification to form each in-context learning (ICL) example. Then, we employ Llama 3.1 70B Instruct model with 4 ICL examples Dubey et al. (2024). The generated specifications are manually evaluated for quality and to ensure they do not mention variable names or counts, parameter values, or constants included in the constraint matrix. For an example of the prompt for generating specifications, see Appendix A.2.

**Including Mathematics in Specifications.** In practice, there are instances of specifications that include mathematical formulations of some of the problem constraints. To emulate this, we also include algebraic constraints formulated in LaTeX and include them in our specifications. To do so, we construct the set of constraint types in the constraint matrix, and supply generic, LaTeX formatted expressions of that each of those constraint types, and include them in the specification. For example,

if the problem includes upper bound constraints, we append the text `The problem has the following constraint type:` $x_i \leq b$, to the end of the specification. For additional details, see Appendix A.2.

### 4.3 WHAT-IF QUESTIONS & MODIFIED CANONICAL FORMS

The final 2 pieces of the MPM task are the what-if question, $\mathcal{Q}$, and modified canonical form, $\mathcal{P}^{\star}$. We generate both using a reverse engineering procedure. Specifically, we begin by heuristically modifying the problems canonical form. For example, we might change a parameter value or a constraint limit. The full suite of possible modifications that can be made correspond to the suite of what-if question types (Section 3.3). This change results in a modified canonical form that serves as $\mathcal{P}^{\star}$.

To generate the what-if question, $\mathcal{Q}$, the corresponds to that change, we utilize LLM inference. Again, we utilize Llama 3.1 70B Instruct, but this time using ICL examples that follow the ReAct paradigm Yao et al. (2023). See Appendix A.3 As before, we begin by creating 15 ReAct-style inputs for 15 of the possible constraint type and what-if question type pairs. Then, we create ICL examples that consist of: i) NL4OPT natural language problem, ii) the corresponding canonical form, and iii) the modified canonical form, $\mathcal{P}^{\star}$. The model is tasked with generating the what-if question, $\mathcal{Q}$, that matches the change in the $\mathcal{P}$. We utilize 3 demonstrations to generate each what-if question. All generated what-if questions are manually evaluated for correctness. Specifically, we ensure that the what-if question accurately reflects the change between the original and modified canonical forms ($\mathcal{P}$ and $\mathcal{P}^{\star}$). We found that 92% of generated what-if questions are of sufficiently high quality to be included in the dataset; the rest are either filtered out or manually modified and included. All generated what-if questions—including low quality generations before human modification—are included in a supplement to the dataset.

### 4.4 DATASET STATISTICS

The WIQOR dataset consists of 1,946 instances of the MPM task. As described above, the creation of each task instance is seeded with an example from NL4OPT. However, MPM instances include generated specifications rather than full descriptions of all variables and constraints; canonical forms may have an increased number of variables; each instance includes a generated what-if question; and the goal is to predict a modified canonical formulation, which matches the generated what-if question. Table 1 provides a detailed breakdown of the dataset across training, development, and test splits, highlighting the number of instances for each split and the distribution of what-if question types. The test split is divided into two parts. The first part, called `Test-Base`, contains 396 data points where the number of decision variables from the source optimization problem $\mathcal{S}^0$ remains unchanged.

The second part, `Test-VarAug` (variables augmented), is created by taking a sample of 100 data points from `Test-Base` and augmenting them by increasing the number of decision variables. The number of additional variables varies across four sets: 5, 10, 20, and 30. Each set contains 100 data points, bringing the total number of data points in `Test-VarAug` to 400, and the total number of data points in the full test split to 796. For each set of additional variable datapoints, we assessed how the model performed as the number of decision variables increased, providing insight into its scalability and adaptability to more complex problem formulations.

The majority of the dataset is composed of *constant change* (CC) and *limit change* (LC) questions. The remaining portion is comprised of more nuanced modifications, such as *constraint direction reversals* (CDR) and *variable interaction modifications* (VIM). Table 2 focuses on the types of constraints impacted by these what-if questions. We note the prevalence of *linear* constraints in the dataset, which dominate the modifications, while ratio constraints and specialized forms such as $xy$ and $xby$ are less frequent.

## 5 EXPERIMENTS

We report performance of LLMs in the Llama 3.1 family and GPT-4 on the WIQOR dataset under the ICL paradigm. Specifically, we experiment with the 8 and 70 billion parameter instruct variants as well as the 34 billion parameter code variant of Llama 3.1 and GPT-4. We include Code Llama in

Table 1: Distribution of data splits and types of *what-if* questions in the WIQOR dataset

| Split type | Count | What-If Question type | Count |
|---|---|---|---|
| Train | 994 | *Constant change* (CC) | 444 |
| Dev | 176 | *Limit change* (LC) | 520 |
| Test-Base | 396 | *Constraint direction reversal* (CDR) | 286 |
| Test-VarAug | 400 | *Variable interaction modification* (VIM) | 315 |

Table 2: Distribution of constraint types in the WIQOR dataset

| Constraint type | Count | Constraint type | Count |
|---|---|---|---|
| *sum* | 324 | *linear* | 765 |
| *lower bound* | 186 | *xby* | 24 |
| *upper bound* | 169 | *xy* | 20 |
| *ratio* | 77 | | |

our experiments since they have been reported to perform well with mathematics as well as symbolic reasoning tasks (Madaan et al., 2022). The models are prompted using few-shot learning, with the demonstrations selected in 2 ways:

- **Random:** exemplars are randomly sampled from a set of human-written examples uniformly at random. Check A.4 for an example prompt; and

- **Similarity:** exemplars are selected such that their corresponding what-if question is semantically similar to the test what-if question.

For exemplars chosen using the Similarity approach, we also incorporate chain-of-thought reasoning (Wei et al., 2023) to outline a step-by-step process. This method helps identify the target constraints being modified by the what-if question and pinpoint the corresponding element to be altered in the canonical formulation matrix. We conduct and present results on both the **Test-Base** (Llama-3.1-8B Instruct, Llama-3.1-70B Instruct, Code-Llama-34B) and **Test-VarAug** (Llama-3.1-70B Instruct) subset of the test split of WIQOR.

**Accuracy metric:** The MPM task involves modifying a formulation and verifying whether the change is correct. To evaluate this, we use the exact match metric, comparing the predicted modified mathematical program in its canonical form with the ground truth $\mathcal{P}^{\star}$. This means that every element in both the predicted matrix and $\mathcal{P}^{\star}$ should match for the prediction to be correct.

## 6  RESULTS & ANALYSIS

The performance statistics for the MPM on the Test-Base split of WIQOR, presented in Table 3, provide key insights into how different In-Context Learning (ICL) strategies affect model accuracy. GPT-4 delivers the best performance, achieving 76.15% accuracy with the Similarity CoT ICL variant, while Llama-3.1 Instruct leads open-source models with 71.6% accuracy.

**Impact of Model Size:** The Llama-3.1 70B Instruct model significantly outperforms both the 8B Instruct and Code Llama models. This suggests that model size has a clear and positive impact on accuracy for this task, as the larger model demonstrates superior generalization abilities to handle modifications in mathematical programs. which This performance improvement is consistent across both the **random** and **similarity** based ICL strategies.

**Effectiveness of ICL Strategies:** The difference in accuracy between the random and similarity-based ICL strategies is most notable in the smaller 8B models. The similarity-based ICL variant improves over random selection by 3% for Llama-3.1 Instruct 8B, showing that selecting examples similar to the test instance benefits smaller models with limited generalization capacity.

**Code Llama's Relative Performance:** Despite having a larger size than the Llama-3.1 8B Instruct, the Code Llama 34B model performs worse than the former model by a margin. This suggests that Code Llama's training paradigm may not be as well-suited for tasks such as MPM.

**Llama-3.1 70B's Robustness:** The minimal performance difference between the random and similarity-based ICL strategies for the Llama-3.1 70B Instruct model indicates that this larger model can handle a more diverse set of examples without requiring curated, similar examples. This robustness suggests that the selection of similar instances is less critical when employing larger models, making the Llama-3.1 70B Instruct a more robust choice for tasks like MPM. Figures 4 and 5 show how does the constraint type being changed through the what-if question and the type of what-if question affects the prediction accuracy of Llama-3.1-70B Instruct model on MPM.

| Model | Model Size | ICL Variant | Accuracy (%) |
|---|---|---|---|
| Llama-3.1 Instruct | 8B | Random | 35.14 |
| Llama-3.1 Instruct | 8B | Similarity CoT | 38 |
| Llama-3.1 Instruct | 70B | Random | 71.6 |
| Llama-3.1 Instruct | 70B | Similarity CoT | 69.69 |
| Code Llama | 34B | Random | 26.42 |
| Code Llama | 34B | Similarity CoT | 33.24 |
| GPT-4 | | Random | 72 |
| GPT-4 | | Similarity CoT | 76.15 |

Table 3: Comparison of ICL strategies for different Llama models of different sizes and training paradigms and GPT-4 on the `Test-Base` split of WIQOR

**Performance on different constraint types:** There's an inverse relationship between constraint complexity and model performance. Simple constraints like *upperbound (ub)* and *lowerbound (lb)* show consistently high accuracy, while more complex ones like *ratio* perform worse. *Linear* constraints maintain high accuracy across both approaches, likely due to their prevalence in optimization problems and training data of the Llama-3.1 70B model. *Ratio* constraints perform poorly in both the random and similarity based ICL variants, indicating this constraint type to be the toughest.

**Performance on different what-if question types:** Questions requiring *variable interaction modifications* (VIM) are the most challenging. The model excels at boundary-related changes, with near 95% accuracy on *limit change* (LC) questions, consistent with high performance on *upperbound* and *lowerbound* constraints.

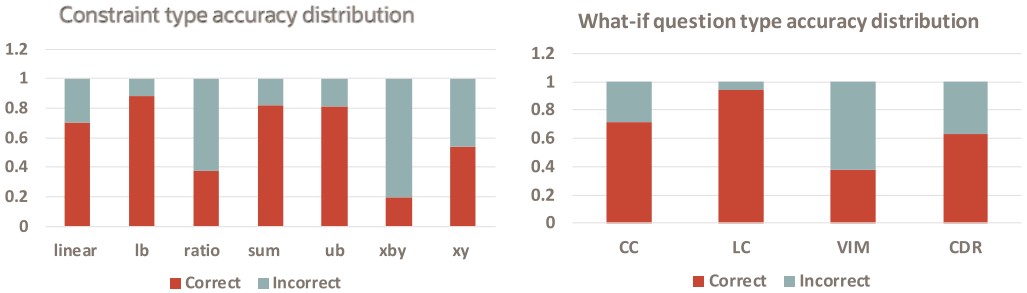

Figure 4: Accuracy distribution with Llama-3.1 70B Instruct using **random** ICL variant

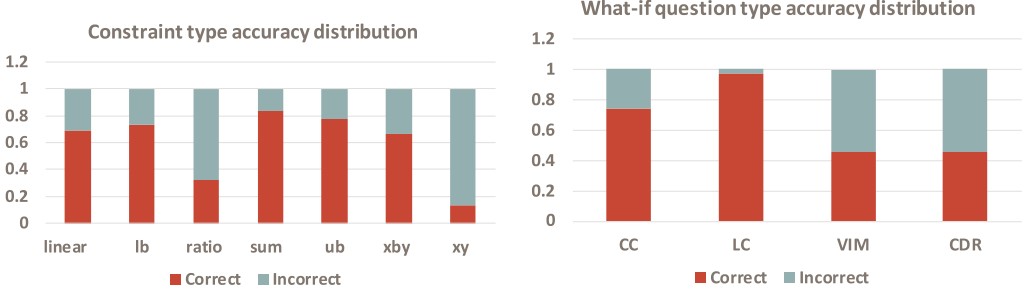

Figure 5: Accuracy distribution with Llama-3.1-70B Instruct using **similarity** CoT ICL variant

**Performance with increasing complexity:** Figure 6 provides valuable insights into the MPM task performance of Llama-3.1-70B Instruct on the `Test-VarAug` split as the complexity of the problems increases. Both **random** and **similarity** based CoT ICL variants show declining performance as the number of decision variables increases, suggesting that the model struggles with increased problem complexity. The similarity CoT ICL consistently outperforms the random ICL across all variable counts. This superiority demonstrates the effectiveness of incorporating step-by-step reasoning and similarity based example selection.

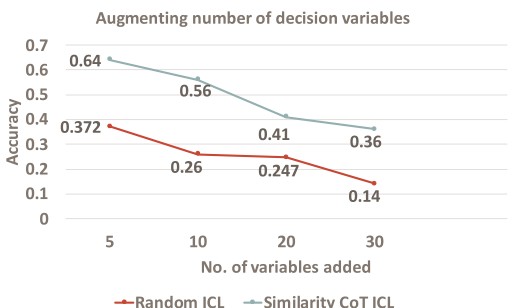

Figure 6: Accuracy v/s number of augmented variables

While both methods show performance degradation, the CoT approach exhibits more resilience. The accuracy drop for CoT (from 0.64 to 0.36) is less steep compared to standard ICL (from 0.372 to 0.14) as variables increase from 5 to 30. However, the performance gap between the two methods narrows as complexity increases, particularly beyond 20 variables. This could indicate a limit to the benefits of CoT as problems become extremely complex, possibly due to the model's inherent limitations or the increased difficulty in maintaining coherent reasoning chains for highly complex problems.

## 7 CONCLUSIONS

We introduce MPM, the task of modifying the canonical formulation of an optimization problem from a problem specification, initial formulation, and a given what-if question. This embodies a type of "what-if" analysis highly valuable to domain experts but often inaccessible. We also present WIQOR, the first dataset for MPM, containing around 2000 problems, 7 constraint types, 4 what-if question types, and varying levels of complexity, measured by the number of decision variables. Our experiments show that the 8B parameter Llama 3.1 model solves only 38% of the simplest WIQOR problems, while the 70B Llama 3.1 instruct model achieves 71.6% accuracy on the simplest cases, but performance drops to 36% as the number of decision variables increases. These results indicate that LLMs have substantial room for improvement before they can effectively assist in real-world industrial what-if analysis, where problem complexity is much higher. Future research could explore hybrid approaches that combine language models with traditional optimization techniques, enhance Chain-of-Thought (CoT) prompting for complex problems, and develop models better suited for handling tasks with numerous variables. In summary, while Llama 3.1-70B Instruct shows promise in handling MPM tasks significant advancements are needed to maintain high performance as problem complexity increases. We hope this work paves the way for techniques that empower domain experts to solve their optimization problems more independently.

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

# A    APPENDIX

## A.1    CONSTRAINT TYPES

You may include other additional sections here.

| Constraint Type | Mathematical representation |
| --- | --- |
| sum | $x + y \leq c$ |
| upperbound | $x \leq c$ |
| lowerbound | $x \geq c$ |
| linear | $a_1 x + a_2 y \leq c$ |
| ratio | $x \leq c(x + y)$ |
| xby | $x \leq ay$ |
| xy | $x \leq y$ |

Table 4: Constraint Types and Their Mathematical Representations as used in Ramamonjison et al. (2023)

## A.2    PROMPT FOR GENERATING SPECIFICATION PROBLEMS

```
 Generate a specification for the given optimization problem,
this specification should be a short summary of the problem
and should not mention the name or number of decision variables
in the problem.  The specification should include the types of
constraints the problem but should not mention the numerical
parameter values about the constraints and the decision variables.
Here are a few examples of optimization problems and their
specifications:

# Example 1
Problem: [optimization problem]
Specification: [specification]

# Example 2
Problem: [optimization problem]
Specification: [specification]

# Example 3
Problem: [optimization problem]
Specification: [specification]

Problem :  You are playing a game where you have to throw a ball
at a target.  Throwing a small ball is worth 5 points and throwing
a large ball is worth 2 points.  You can throw at most 20 balls
```

total. You must also throw at least 6 small balls and 5 large
balls. You cannot throw more than 12 of either type. Assuming
you always hit the target, how many balls of each type should you
throw to maximize your score? What is that score?

**Resultant specification:** You are playing a game where you have to throw a
ball at a target. You score some number of points for throwing a
given type of ball. There is a limit on the total number of balls
that you can throw. There is also a lower and upper limit on the
number of each type of ball you can throw. How many balls of each
type should you throw to maximize your score?

### A.2.1 INCLUDING MATHEMATICS IN SPECIFICATIONS

The specifications also have the LaTeX representation of the constraint types found in the
problem. These representations are generated heuristically based on the type of constraints.
Refer to the table for the mapping between each constraint type and its corresponding LaTeX form.

| Constraint Type | Mathematical Form | LaTeX Text (used in prompts) |
|---|---|---|
| sum | $\sum_i a_i \times x_i \leq b$ | `\sum_{i} a_i \times x_i \leq b` |
| upperbound | $x_i \leq b$ | `x_i \leq b` |
| lowerbound | $x_i \geq b$ | `x_i \geq b` |
| linear | $\sum_i a_i \times x_i \leq b$ | `\sum_{i} a_i \times x_i \leq b` |
| ratio | $x_i \leq c \times \sum_i x_i$ | `x_i \leq c \times \sum_{i} x_i` |
| xby | $x_i \leq b \times y$ | `x_i \leq b \times y` |
| xy | $x_i \leq y$ | `x_i \leq y` |

### A.3 PROMPT FOR GENERATING WHAT-IF QUESTIONS

**Instruction:** Using the problem description, the original
constraints, and the modified constraints, identify the specific
change made between the two sets of constraints and generate a
"what-if" question that reflects this modification. Ensure that
the question describes the altered condition in the context of the
original problem, focusing on the entity or parameter affected by
the change.

**Problem:** A hotel employs cleaners and receptionists. Cleaners
earn $500 per week and receptionists earn $350 per week. The
hotel requires a minimum of 100 workers of whom at least 20 must
be receptionists. To keep the hotel clean and running smoothly,
the number of receptionists should be at least a third of the
number of cleaners. The hotel wants to keep the weekly wage bill
below $30,000. Formulate a linear program (LP) to minimize the
wage bill.

**Order Mapping:** [cleaners:0, receptionists:1]

**Constraints:** Original Constraints : [[-1.0, -1.0, -100.0], [0.0,
-1.0, -20.0 ], [0.3333, -1.0 , 0.0] , [500.0, 350.0 , 30000.0]]

Modified Constraints :[ [-1.0, -1.0, -105.0], [0.0, -1.0, -20.0],
[0.3333, -1.0, 0.0], [500.0, 350.0, 30000.0]]

Objective : [500.0,350.0]

**React Prompt:**
  Act 1: Think[I need to analyze the differences between the original
and modified formulations to identify the change.]
  Act 2: Compare constraints
  Obs 2: The first constraint changed from [-1.0, -1.0, -100.0] to

```
-1.0, -105.0!
.
   Act 3:   Think[Let's analyze this constraint in detail to understand its
meaning.]
   Act 4:   Interpret constraint format
   Obs 4:   The constraint is in the form [a, b, c], where 'a' corresponds
to cleaners, 'b' to receptionists, and 'c' is the right-hand side of the
inequality.
   Act 5:   Analyze coefficients
   Obs 5:   The coefficient for cleaners (a) is -1.0, and for receptionists
(b) is -1.0.
   Act 6:   Interpret inequality
   Obs 6:   Since the constraint is in a less than or equal form, the
negative values for 'a', 'b', and 'c' indicate a 'greater than or equal to'
constraint when converted to standard form.
   Act 7:   Formulate constraint equation
   Obs 7:   The constraint [ -1.0, -1.0, -100] translates to cleaners +
receptionists ≥ 100, and the modified constraint [ -1.0, -1.0, -105]
translates to cleaners + receptionists ≥ 105.
   Act 8:   Think[This constraint specifies the minimum number of total
workers, which has increased from 100 to 105.]
   Act 9:   Formulate a what-if question
   Obs 9:   What if the hotel decided to increase the minimum number of
workers to 105?
   Act 10:  Verify impact on formulation
   Obs 10:  Changing the minimum number of total workers to 105 modifies
the first constraint.
   Act 11:  Finalize the what-if question
   Obs 11:  The what-if question is accurate.
```

**Resultant What-if Question**: What if the hotel decided to increase
the minimum number of total workers to 105?

A.4  PROMPT FOR GENERATING MODIFIED FORMULATIONS (THE MPM TASK)

```
Imagine you are a combinatorial optimization problem solver.  I
will give you an optimization problem which has specifications
about the problem's domain and a brief description of the
constraints and the target optimization.  You will also be given
the order mapping for columns of constraints matrix to variables.
In cases where there are multiple constraints of the same type
you will be given an entity to constraint mapping (row index (zero
indexed) of the constraints matrix) to disambiguate the entities
being constrained by the rows having the same constraint types.
In addition to this a what-if question will be given to you, using
which you have to generate the modified canonical formulation for
the same problem.  Only output the modified canonical formulation
in matrix form and output nothing else.
```

Here are some examples:

**Example 1**

**Specification problem**: A hotel employs some types of workers.
Both types have a fixed wage.  There is a minimum number of total
workers required, out of which at least a given number must be a
specific type of worker.  There is a lower bound on the ratio of
the number of each type of worker and an upper bound on the total
wage.  Formulate an LP to minimize the wage bill.

**Order mapping**: [cleaners:0, receptionists:1]

Constraint to entity mapping:

**Formulation**: Constraints: [[-1.0, -1.0, -100.0], [-0.0, -1.0, -20.0], [0.333333333333333, -1.0, -0.0], [500.0, 350.0, 30000.0]] Objective: [500.0, 350.0]

**What-if question**: What if the hotel decided to increase the minimum number of total workers from 100 to 105?

**Modified formulation**: Constraints: [[-1.0, -1.0, -105.0], [-0.0, -1.0, -20.0], [0.333333333333333, -1.0, -0.0], [500.0, 350.0, 30000.0]] Objective: [500.0, 350.0]

**Example 2**

**Specification problem**: You are playing a game where you can play some kinds of shots and each kind is worth some number of points. There is a limit on the total number of shots that you can take. There is also a lower limit on the number each type of shot that you must take. There is an upper limit on the number of shots of each type that you can take. How many of each shot must you take, assuming all your shots get points, to maximize your score?

**Order mapping**: [short shots:0, long shots:1]

Constraint to entity mapping:

**Formulation**: Constraints: [[1.0, 1.0, 14.0], [-1.0, 0.0, -5.0], [-0.0, -1.0, -2.0], [1.0, 0.0, 8.0], [0.0, 1.0, 8.0]] Objective: [2.0, 5.0]

**What-if question**: What if the number of short shots allowed was 6 instead of 8?

**Modified formulation**: Constraints: [[1.0, 1.0, 14.0], [-1.0, 0.0, -5.0], [-0.0, -1.0, -2.0], [1.0, 0.0, 6.0], [0.0, 1.0, 8.0]] Objective: [2.0, 5.0]

**Example 3**

**Specification problem**: A retired professor wants to invest some capital in some industries. Each dollar invested in an industry yields a fixed profit. There is a lower limit on the amount that must be invested in one of the industries and a lower limit on the ratio of the amount invested in one of the industries to the total amount invested. Formulate an LP that can be used to maximize the professor's profit.

**Order mapping**: [airline:0, railway:1]

Constraint to entity mapping:

**Formulation**: Constraints: [[1.0, 1.0, 50000.0], [-0.0, -1.0, -10000.0], [-0.75, 0.25, -0.0]] Objective: [0.3, 0.1]

**What-if question**: What if the professor decided to decrease the minimum percentage of investment in the airline industry from 25% to 20% of the total investment?

**Modified formulation**: Constraints: [[1.0, 1.0, 50000.0], [-0.0, -1.0, -10000.0], [-0.80, 0.20, -0.0]] Objective: [0.3, 0.1]

Now here is a specification problem, it's original formulation and a what if question for you to output a modified formulation.

Output only the updated constraints and objectives and nothing else

**Specification problem:** A macro-counting fitness guru only eats some types of meals. Each type of meal has a fixed amount of calories, protein, and sodium. There is a lower bound on the total calories and protein that the guru needs to eat. There is also an upper bound on the ratio of the number of one type of meal to the total number of meals. How many of each type of meal should he eat to minimize his sodium intake?

**Order mapping:** 'salmon': 0, 'eggs': 1

Constraint to entity mapping: $'total_calories' : 0, 'total_protein' : 1$

**Formulation:** Constraints: [[-300.0, -200.0, -2000.0], [-15.0, -8.0, -90.0], [-0.4, 0.6, 0.0]] **Objective:** [80.0, 20.0]

**What-if question:** What if the fitness guru decided to increase his minimum protein requirement to 108 grams?

## A.5 VARIABLE AUGMENTATION ALGORITHM

---

**Algorithm 1** Variable Augmentation Algorithm

---

1: **Input:** $p\_idx$: idx of diff row between $P$ and $P^*$, $cf$: canonical form, $wiq\_type$: what-if question type
2: **Output:** $aug\_cf$: augmented canonical form matrix
3: $aug\_cf \leftarrow []$
4: **for** $idx, row \in cf$ **do**
5:     $constraint\_type \leftarrow get\_constraint\_type(row)$
6:     $limit \leftarrow row[-1]$
7:     $coeff\_row \leftarrow row[:: -1]$
8:     $aug\_row \leftarrow coeff\_row$
9:     $aug\_rows \leftarrow []$
10:     **if** $constraint\_type == sum$ **then**
11:         $row\_ele \leftarrow row[0]$
12:         **if** $p\_idx == idx$ **then**
13:             **if** $wiq\_type == limit\_change$ **then**
14:                 $coeff\_row.append(row\_ele)$
15:                 $coeff\_row.append(limit)$
16:             **else if** $wiq\_type == vim$ **then**
17:                 $coeff\_row.append([0] * aug\_k)$
18:                 $coeff\_row.append(limit)$
19:                 $aug\_row.extend([row\_ele] * aug\_k)$
20:                 $aug\_row.append(limit * 1.5 * aug\_k)$
21:             **else if** $wiq\_type == cdr$ **then**
22:                 $coeff\_row.extend([row\_ele] * aug\_k)$
23:                 $coeff\_row.append(limit)$
24:             **end if**
25:         **else**
26:             $coeff\_row.extend([0] * aug\_k)$
27:             $coeff\_row.append(limit)$
28:         **end if**
29:     **else if** $constraint\_type == linear$ **then**
30:         $row\_ele \leftarrow mean(coeff\_row)$
31:         **if** $p\_idx == idx$ **then**
32:             **if** $wiq\_type == limit\_change$ **then**
33:                 $coeff\_row.append(row\_ele)$
34:                 $coeff\_row.append(limit)$
35:             **else if** $wiq\_type == constant\_change$ **then**
36:                 $coeff\_row.append(row\_ele)$
37:                 $coeff\_row.append(limit * 1.5 * aug\_k)$
38:             **else if** $wiq\_type == vim$ **then**
39:                 $coeff\_row.append([0] * aug\_k)$
40:                 $coeff\_row.append(limit)$
41:                 $aug\_row.extend([row\_ele] * aug\_k)$
42:                 $aug\_row.append(limit * 1.5 * aug\_k)$
43:                 $aug\_rows.append(aug\_row)$
44:             **else if** $wiq\_type == cdr$ **then**
45:                 $coeff\_row.extend([row\_ele] * aug\_k)$
46:                 $coeff\_row.append(limit)$
47:             **end if**
48:         **else**
49:             $coeff\_row.extend([row\_ele] * aug\_k)$
50:             $coeff\_row.append(limit * 1.5 * k)$
51:         **end if**
52:

---

**Algorithm 2** Variable Augmentation Algorithm (Continued)

---

1: **for** $idx, row \in cf$ **(continued) do**
2:     **if** $constraint\_type == lowerbound$ **or** $constraint\_type == upperbound$ **then**
3:         $coeff\_row.extend([0] * aug\_k)$
4:         $coeff\_row.append(limit)$
5:         $aug\_row \leftarrow [0] * len(coeff\_row)$
6:         $aug\_row.extend([0] * aug\_k)$
7:         $aug\_row.append(limit)$
8:         **for** $i \in range(aug\_k)$ **do**
9:             $aug\_row\_copy \leftarrow copy(aug\_row)$
10:           $aug\_row\_copy[len(coeff\_row) + i] \leftarrow -1$ **if** $constraint == lowerbound$ **else** $1$
11:           $aug\_rows.append(aug\_row\_copy)$
12:         **end for**
13:     **else if** $constraint\_type == ratio$ **then**
14:         $ratio\_ele \leftarrow check\_ele\_not\_one(coeff\_row)$
15:         $coeff\_row.extend([ratio\_ele] * aug\_k)$
16:         $coeff\_row.append(limit)$
17:     **else if** $constraint\_type == xby$ **then**
18:         $coeff\_row.extend([0] * aug\_k)$
19:         $coeff\_row.append(limit)$
20:     **else if** $constraint\_type == xy$ **then**
21:         $coeff\_row.extend([0] * aug\_k)$
22:         $coeff\_row.append(limit)$
23:     **end if**
24:     $aug\_cf.append(coeff\_row)$
25:     $aug\_cf.extend(aug\_rows)$
26: **end for**
       **return** $aug\_cf$ =0

---

