# OpenReview forum: "WIQOR: A dataset for what-if analysis of Operations Research problems"
_ICLR.cc/2025/Conference — Submitted to ICLR 2025_

### Official Review · Reviewer_mpi3 · 2024-10-29

**Soundness:** 2
**Presentation:** 1
**Contribution:** 2
**Rating:** 5
**Confidence:** 3

**Summary:**

- This paper proposed a new task, mathematical program modification (MPM), whose goal is to revise a math program (MP) based on a natural language instruction.
- The paper presented a dataset WIQOR, of 1946 MPM instances, derived from NL4OPT dataset. WIQOR contains 4 set, train, dev, test-base and a harder test-VarAug. To build the dataset, the authors first modify the original MP with 4 potential ways (LC, CC, CDR, VIM, explained in paper), and then prompt LLMs, as a reverse engineering approach, to generate the **What-if questions** based on the difference between the original MP and the modified MP.
- The author test 3 variants of Llama model on the proposed dataset and the experiment results show it is a challenging dataset.

**Strengths:**

- The collected dataset could be a contribution to the community, however if it is actually emulating the real-life scenario is still questionable.
- It is interesting to see how different constraint types and what-if question types pose different levels of challenges for LLMs.

**Weaknesses:**

- This paper is clearly a rushed work. The presentation is draft-like ( For example, the margin of Figure 6, the unfinished appendix A1,2,3), the writing is a bit verbose ( for example Section 2 related work and Section 3.1 & 3.2).

- The evaluation section is limited.
    - As shown in Table 3, the dataset is only evaluated on very few models, Llama-3.1-8B/70B and a code-Llama.
    - When comparing ICL variant, zero-shot performance is not tested.
    - When comparing CoT performance, ablation study should be conducted to compare the performance with and without CoT.
    - As very few models are evaluated, the conclusions drawn in Section 6 is not well supported.

- The motivation for the proposed dataset is not convincing enough, and the scope of the project is relatively small.
    - The whole motivation of the dataset is built on the assumption that in real-life people will communicate/modify a MP in such a way, but your specification S is just summary and do not contain detailed information. Why do you think that's the actual way people communicate?
    - The proposed modification approach should be compared with a NL-to-MP baseline, where the original optimization text + modifications is given to the LLM to generate a new MP. If this actually works better than modifying the existing MP, I don't see the motivation for this approach.
    - How good is the coverages of What-if questions? Do they cover all potential aspects of a MP problem? Are all the MP problems linear programming problems? You can include a limitation section to discuss the coverage of the What-if questions.

**Questions:**

- Are all the MP questions included Linear programming? If not, did you treat linear and non-linear differently or did you investigate their performance difference?
- What's the actual prompt you used to evaluate the performance of Llama-3.1? Did you include specification in your prompt? Please show a template example.
- You specification S does not contain specific description about the variables. Does it mean after the modification, the specifications are still guaranteed to be valid?
- What's the motivation for rewriting the optimization problem to the specifications?

---

> ### Author Response · Authors · 2024-11-29
> **Response to Reviewer mpi3 Comments**
>
> Thank you for taking the time to review our work. We appreciate the strengths you’ve highlighted, particularly the contribution of our dataset to the community. We’re glad to see that the varying challenges posed by different constraint and what-if question types have been recognized as valuable for evaluating LLMs. Below, we provide clarifications and responses to your questions and comments.
>
> ### Weakness 1
> We appreciate the feedback and understand the concerns raised. While we made every effort to ensure the clarity and quality of the paper, we recognize that some aspects may not have met the standards expected.
>
> - We have addressed minor issues such as the margin problem in Figure 6 and have refined the formatting of the Appendix. Although the appendix was complete in itself, we have added more sections and improved the overall presentation.
> - We have made the Related Work section more concise. As for Sections 3.1 and 3.2, these were carefully written to explain the definitions and motivations behind specifications and what-if questions. Our intention was to keep them focused and direct, and we apologize if they seemed overly detailed or verbose.
>
> ### Weakness 2
>
> -  **As shown in Table 3, the dataset is only evaluated on very few models: Llama-3.1-8B/70B and Code-Llama.**
> To address this concern, we have expanded our evaluations to include GPT-4. GPT-4 was tested using both random few-shot exemplar (without CoT) selection and similarity-based CoT settings. The results showed consistent trends with those observed for the Llama-3.1 models, though GPT-4 demonstrated superior overall performance. These new findings, along with detailed performance statistics, have been added to Table 3 of the revised paper and are also given below.
>
> | Model   | Setting                        | Accuracy (%) |
> |---------|--------------------------------|--------------|
> | GPT-4   | Zero shot                     | 63.15        |
> | GPT-4   | Few shot                      | 72.00        |
> | GPT-4   | Few shot CoT                  | 76.76        |
> | GPT-4   | Random few shot - 5 augmented variables | 44.44        |
> | GPT-4   | Similarity few shot CoT - 5 augmented variables | 54.83        |
>
> -  **Zero-shot performance is not tested when comparing the ICL variant.**
> Zero-shot performance was not evaluated in the original submission because the models struggled to conform outputs to the expected matrix format without few shot examples. The performance was notably poor, which we believed for brevity reasons would not contribute meaningfully to the analysis. We have presented zero shot results with GPT-4 in the previous point for completeness.
>
> -  **For CoT performance, an ablation study should be conducted to compare performance with and without CoT.**
> We conducted evaluations under two main settings (Section 5)—random few shot exemplar selection (without CoT) and similarity-based CoT—and present the detailed performance statistics and analysis for different models in Section 6.
>
> - **Conclusions drawn in Section 6 are not well-supported due to limited evaluations.**
> To strengthen the conclusions in Section 6, we have included additional results from our expanded evaluations, particularly with GPT-4. The consistency of trends across Llama-3.1 models and GPT-4 supports the claims made in the paper. We hope these added results address concerns regarding the support for our conclusions and enhance the overall robustness of our findings.
>
> ### Weakness 3
>
> -  **The whole motivation ..**
> We appreciate the critique and would like to clarify that our approach reflects established real-world practices. Specifications, while concise, work alongside the initial formulation to form a complete problem instance for posing and answering what-if questions.
>
> Specifications summarize key elements—such as variable groups, relationships, and constraints—without detailing every parameter or variable (Section 3.1,[1]). In practice, these details are stored in databases or chosen through user interfaces and are incorporated into the initial formulation by engineers. Together, the specification and formulation form a comprehensive problem instance for a what-if scenario.
> The what-if scenario question is never the first interaction between a user and an engineer. By this stage, the user is already aware of the initial numerical settings and overall formulation, allowing them to meaningfully explore how changes might impact the problem without needing to restate every detail.
>
> In our MPM task (Section 3), users begin with this complete representation of a what-if question. The specification provides a natural language overview, and the formulation includes precise mathematical details. This combined information ensures that users can explore what-if scenarios without redefining the entire problem.
> By structuring our dataset in this way, we reflect the iterative nature of real-world workflows which includes handling of what-if scenarios ([2]).

---

> ### Author Response · Authors · 2024-11-29
> **Response to Reviewer mpi3 Comments**
>
> ### Weakness 3(cotd.)
>
> - **The proposed modification approach should be compared with a NL-to-MP baseline**
>
> We appreciate the suggestion to compare our approach with an NL-to-MP baseline. However, counterfactual analysis of an existing problem and formulation aligns better with real-world optimization workflows, where problems are rarely defined in natural language (NL). Instead, they are typically communicated through specifications[1] paired with initial formulations, which form a more complete and practical representation of the what-if problem.
>
> Furthermore, what-if exploration is inherently iterative. Users frequently refine their understanding of a problem by posing multiple questions on the same formulation (Section 1). Modifying an existing MP for each question ensures continuity and consistency across iterations. Regenerating the MP from scratch for every what-if question, as proposed, would be computationally inefficient and risk inconsistencies, especially when modifications build on earlier changes.
>
> In conclusion, while an NL-to-MP baseline may provide an academic comparison, our approach is more scalable and closely mirrors how what-if exploration of operations research problems is actually done [2].
>
> - **How good is the coverages of What-if questions?**
>
> We appreciate the inquiry regarding the coverage of what-if questions and the scope of modifications included in our dataset. Our work focuses exclusively on linear programming problems, and the types of what-if questions we consider are explicitly outlined and detailed in Section 3.3. These categories represent common modifications to individual constraints in real-world optimization scenarios.
>
> While we acknowledge that there are many possible what-if scenarios, it is nearly impossible to account for every conceivable variation. Our study deliberately focuses on four specific types of modifications: limit changes, constant changes, constraint direction reversals, and variable interaction modifications. These categories were selected because they capture the majority of practical modifications users make to constraints during iterative problem-solving. Questions that involve altering interactions between multiple constraints or introducing conditional dependencies fall outside the scope of this work.
>
> We hope this clarification sheds more light on the aspects modified by our what-if questions and their coverage. If there is a more quantifiable coverage metric that the reviewer is looking for, we are happy to evaluate that.
>
> ### Questions
>
> - **Question 1.**
> Yes, all the MP questions in our dataset are linear programming problems. As already explained in the paper, this is because our dataset builds upon the NL4Opt dataset (Section 4), which exclusively contains linear programming problems.
>
> - **Question 2.**
> Thank you for your question. We have included an example of the random few-shot exemplar prompt in Appendix Section A.4. Yes, the prompt does include the specification, as it is a key component of the input for the MPM task (defined in Section 3), alongside the original formulation.
>
> - **Question 3.**
> It is true that the specifications do not include numerical details about the variables, and it is possible for the specifications to no longer align perfectly after a modification. However, this is not because of the absence of variable-specific descriptions; the variables (their number and names) themselves remain unchanged after modifications. There might be changes to the parameter values(price of petrol in city X) of the variables, but this never renders the specification invalid.
>
> For example, in the case of a "constraint direction reversal" modification, an upper bound constraint might be converted into a lower bound constraint, or vice versa. Such changes affect the type of constraints but do not impact the variables or their definitions. Moreover, the validity of the specification after a modification is not a concern, as the focus is on accurately reflecting the modifications in the formulation, not on whether the specification remains perfectly aligned.
>
> - **Question 4.**
> As explained in Sections 3.1 and 4.2, the motivation behind rewriting the optimization problem into a specification helps us mirror real-world optimization workflows, where problems often involve hundreds or thousands of variables and constraints. Rather than overwhelming users with every numerical detail, specifications distill the key elements of the problem, such as variable groups and constraint relationships making the problem more interpretable and communicable. It also allows us to easily scale our problem to more variables.
>
> .[1] Optimization modeling and verification from problem specifications using a multi-agent multi-stage llm framework. INFOR: Information Systems and Operational Research,pp. 1–19, 2024
>
> [2] Large Language Models for Supply Chain Optimization Beibin Li, Konstantina Mellou, Bo Zhang, Jeevan Pathuri, Ishai Menache

---

> ### Author Response · Authors · 2024-12-02
> **Request for review of our clarifications**
>
> Dear reviewer mpi3, Since today is the last day for posting messages to the authors, we kindly request you to review our complete and detailed clarifications. If you have any remaining questions, please feel free to seek clarification. If our responses address your concerns adequately, we would sincerely appreciate your consideration in revising the assessment scores for our submission.
>
> Thank you, the Authors

---

### Official Review · Reviewer_6sCn · 2024-11-01

**Soundness:** 2
**Presentation:** 3
**Contribution:** 2
**Rating:** 3
**Confidence:** 5

**Summary:**

This paper develops a novel dataset, using the dataset from NL4OPT as a seed to expand the original dataset. The complexity of generated problems, as well as the number of variables and constraints, is gradually increased to obtain more complex data. The provided canonical form is correctly modified in response to the what-if question.

**Strengths:**

- The paper proposes a self-instruction-based approach to synthesize complex reasoning data, using a progressive method to generate more complex problems and solutions.

- The article is well-written, and the method for expanding the dataset is simple and clear.

- It explores some interesting combinations of conditions and variables, which is very helpful for investigating the boundaries of linear programming problems.

**Weaknesses:**

- The novelty of the data generation method is limited[1,2]. I believe that the self-instruction-based method may face serious mode collapse[3] issues, and it’s necessary to discuss the presence of this problem in mathematical problem synthesis. As for improvements, I think a human-in-the-loop approach could be provided for optimization.
- Using exact match to calculate accuracy is unreasonable; why not provide results on the solver instead? Exact match can be overly restrictive and often misleading in evaluating mathematical and reasoning tasks, as it treats answers as binary (correct or incorrect) without considering near-correct solutions that may only have minor calculation differences. Real-world problem-solving often requires that a solution meet certain functional criteria, like feasibility within constraints or optimality, rather than matching a single predetermined answer exactly. Using solver-based evaluations provides a nuanced assessment, confirming if solutions meet problem requirements under real conditions. Solvers can verify the feasibility and robustness of a solution within the problem’s constraints, accounting for slight variances and rounding errors that exact match might unfairly penalize. This approach aligns the evaluation with real-world performance, offering clearer insights into the model's practical applicability and helping to identify areas for meaningful improvement. By assessing solutions based on their functionality and effectiveness, solver-based evaluation ultimately yields a more refined measure of a model's problem-solving capability, moving beyond rigid surface-level accuracy.
- Can it be extended to more complex mixed-integer programming problems? Mixed-integer programming introduces additional complexities compared to standard linear programming due to the presence of integer constraints, which often lead to a more challenging solution space and can significantly increase the computational difficulty of finding optimal solutions. For example, extending this method to MIP might require adjustments in handling discrete variables, which can limit certain relaxation techniques commonly applied in continuous optimization problems. Additionally, it would be important to investigate how the current framework could address or adapt to the combinatorial nature of integer constraints. This could involve rethinking the approach to feasible region exploration, as well as adapting constraint handling mechanisms to account for mixed integer domains. Overall, this extension could open up new application avenues, and exploring the specific requirements for such an adaptation might reveal valuable insights into the method's versatility and practical scalability across diverse optimization scenarios.
- I haven't seen any code or datasets; I would like to carefully check whether the data questions truly align with real-world solving objectives.
- I believe it is essential to provide the solvability rate of the generated problems and to include test results across different solvers, comparing their performance.

**Reference**
[1] Self-Instruct: Aligning Language Models with Self-Generated Instructions.
[2] Wizardlm: Empowering large language models to follow complex instructions.
[3] Is Model Collapse Inevitable? Breaking the Curse of Recursion by Accumulating Real and Synthetic Data

**Questions:**

- I believe many experimental results are missing. For instance, have you evaluated whether the generated solution formulas are solvable? Additionally, numerous generated problems need to be tested on the solver.
- How is data quality ensured, given that many problems generated by the model often do not exist in the real world? Are there any automatic checks or manual evaluation steps to filter out unrealistic problems? I believe that the validity of the generated problems needs to be supported by detailed experimental results to be convincing.
- How is data diversity ensured, considering that simple semantic similarity is typically not a good metric for mathematical reasoning tasks?  I believe it is necessary to provide some metrics or analysis to demonstrate the diversity of the dataset and to ensure that mode collapse has not occurred.

---

> ### Author Response · Authors · 2024-11-28
> **Response to Reviewer 6sCn Comments**
>
> Thank you for your thoughtful and detailed review. We appreciate your recognition of the strengths of our work, including the approach to synthesize complex reasoning data, the simplicity and clarity of our progressive dataset expansion method, and the exploration of interesting combinations of conditions and variables to investigate the boundaries of linear programming problems.
> Below, we provide clarifications and responses to your questions and comments.
>
> ### Weakness 1
>
> **Novelty of the data generation method:**
> We appreciate your thoughts but find some aspects of this statement unclear. Specifically, it’s not clear which dataset component’s generation method you are referring to regarding the lack of novelty. To clarify, our goal was not to introduce a novel data generation strategy but to ensure accurate and high-quality generation of dataset components.
>
> **Mode collapse:**
> We assume you are referring to "mode collapse" rather than "model collapse," as discussed in [3] above, since model collapse typically arises when a model is iteratively trained on its own generated data, which is not the case in our work. Our data generation method combines heuristics with human evaluation. Specifically, target formulation matrices are created by systematically modifying the original matrices through heuristics, and the corresponding what-if questions are generated using these matrices via carefully constructed ReAct prompts (Appendix Section A.3) for Llama 3.1 70B Instruct. These undergo a rigorous human evaluation, and the specification problems are also carefully reviewed before inclusion in the final dataset (Section 4.2). Necessary changes are made to these components during the evaluation to correct inaccuracies.
> To assess the diversity of the generated data, we calculated the Self-BLEU score ([1]) for the what-if questions across the entire dataset, obtaining a value of 0.50156, which suggests that they are not highly similar.
> Moreover, the what-if questions in our dataset are short, simple, and objective, such as, "What if the price of this product was $50?", reducing room for degenerate outputs. Additionally, the specification problems, as explained in the paper, are concise summaries of the NL4Opt optimization problems, leaving minimal room for ambiguous or overly repetitive patterns.
>
> **Optimization:**
> If the 'optimization' you suggest pertains to a component of the data generation strategy, its overall quality can be assessed through our human evaluation results. As mentioned in the paper (Section 4.3), the accuracy of the what-if questions generated was 92%, which reflects the robustness of our approach. We hope this addresses your concerns and are happy to clarify further if needed.
>
> [1] Texygen: A benchmarking platform for text generation models.
>
> ---
>
> ### Weakness 2
>
> **“Using exact match to calculate accuracy is unreasonable; why not provide results on the solver instead?”**
> To clarify, our task is not about measuring a model’s ability to formulate or solve an optimization problem. Instead, our dataset evaluates whether a model can accurately identify and implement specific changes in a given mathematical formulation in response to a what-if question. For this task, exact match is the most reasonable evaluation metric because it directly measures whether the required modifications are correct. The focus here is on accuracy in modifying formulations, not solving them.
>
> **“Exact match can be overly restrictive and often misleading...”**
> This argument is more relevant for tasks involving free-form problem-solving or solution synthesis. However, in our task, exact match is entirely appropriate because the goal is to verify precise modifications, not to evaluate correctness in the context of a solver's criteria.
>
> **Regarding solver-based evaluations**:
> While solvers offer an additional verification layer, they depend on correctly formulated inputs. If the formulation is incorrect, evaluating solver performance becomes secondary. Our exact match evaluation directly measures modification accuracy, which is more appropriate for this task.
>
> As for concerns about **“slight variances and rounding errors,”** these are not relevant since our task does not involve calculations. It focuses on modifying specific components of a formulation matrix based on the what-if question. For example, a question like, *“What if the number of packages a truck could transport in a day was 50?”* explicitly identifies the variable and target value. Errors may stem from hallucinations not rounding errors.
>
> **“Solver-based evaluation ultimately yields a more refined measure of a model's problem-solving capability”**:
> This statement conflates problem-solving with formulation modification. Solver evaluation is about feasibility, while our task ensures correct modifications. Exact match directly measures whether the requested changes are made correctly, which is the focus of our task, not solution feasibility.

---

> ### Author Response · Authors · 2024-11-28
> **Response to Reviewer 6sCn Comments**
>
> ### Weakness 3
>
> **“Can it be extended to more complex mixed-integer programming (MIP) problems?”**—if ‘it’ refers to our dataset generation strategy, the answer is yes. The components of our strategy, such as reverse engineering of what-if questions and specification generation, are problem-type agnostic and can be applied to MIP problems with appropriate adjustments.
>
> However, much of the statement seems to discuss optimization problem-solving rather than the task we focus on, which is the modification of existing mathematical programs. We are not developing or proposing a “method” for solving optimization problems. As such, concerns about the "challenging solution space," "feasible region exploration," or "method's versatility and practical scalability across diverse optimization scenarios" are not directly relevant to our contribution.
>
> That said, our dataset and task serve as an initial step toward enabling counterfactual analysis across different types of mathematical programming problems. While the work primarily focuses on linear programming, it lays the groundwork for extending these ideas to more complex problem types, including MIP, in future research. Exploring these extensions could provide insights into the adaptability and applicability of our approach.
>
> ### Weakness 4
>
> Examples of the specification problems and what-if questions in our dataset are provided in the paper in multiple figures (Figures 1 and 3). Upon acceptance, we plan to release the full dataset for transparency. In the revised version, we include more examples in Appendix Section A.2 and A.4.
>
> ### Weakness 5
>
> Thank you for your suggestion, however, we maintain that evaluating the solvability of the problems or including test results across different solvers is not relevant to our work, as the focus of our approach lies elsewhere. Our dataset builds upon the NL4Opt dataset, a widely used and expert-verified resource in operations research and NLP. The specification problems in our work are concise representations of NL4Opt problems, and their validity and solvability were established during the construction of the source dataset.
> The core objective of our work and the MPM task is not to solve optimization problems but to assess whether models can accurately identify and make the required modifications based on what-if questions. We hope this clarifies the distinction between our focus and the proposed evaluation.
>
> ### Questions
>
> ### **Question 1.**
>
> Thank you for your question. As clarified above, our work focuses on assessing whether models can accurately identify and implement modifications based on what-if questions, not on evaluating the solvability of generated formulas. To reiterate, our dataset is derived from the expert-verified NL4Opt dataset, ensuring the validity of the problems. Testing solver performance is outside the scope of this study.
>
> ### **Question 2.**
>
> **“Many problems generated by the model often do not exist in the real world”**
> It is important to note that research datasets, including benchmarks like GSM8K (mathematical reasoning) often contain hypothetical scenarios designed to test specific reasoning capabilities. Similarly, our source dataset, NL4Opt, has textbook-like problems that might never 'exist' in the real world. The aim of these datasets is to test reasoning capabilities; the problems might not be extant in order to do that. Our focus, on the other hand, is to make sure that the format in which a problem is presented mirrors the real world as much as possible, with the specification problems and an increased number of variables, even though the context of the problem might not exist in the real world.
>
> Furthermore, the problems in NL4Opt are expert-verified, and our dataset, because of having been derived from this, preserved this rigor. To add, the specifications and the what-if questions introduced in our work undergo manual evaluation, ensuring both clarity and validity. If by “unrealistic” you mean some other specific aspect, we would appreciate clarification.
>
> ### **Question 3.**
>
> **"How is data diversity ensured?"/"Ensure that mode collapse has not occurred"**
> Our dataset incorporates four distinct types of what-if questions spanning seven kinds of constraints, ensuring a varied set of problem modifications. Additionally, the source dataset, NL4Opt, provides problems from three domains in the training and development sets. For the test set, three additional domains are introduced to further enhance diversity across problem contexts. To assess the diversity of the generated data, we calculated the Self-BLEU score for the what-if questions across the entire dataset, obtaining a value of 0.50156, which suggests that they are not highly similar.
> As explained in clarification to Weakness 1, our data generation process combines heuristic-based modifications with rigorous human evaluation of both the what-if questions and the specification problems to ensure quality and diversity.

---

> ### Author Response · Authors · 2024-12-02
> **Request for review of our clarifications**
>
> Dear reviewer 6sCn,
> Since today is the last day for posting messages to the authors, we kindly request you to review our complete and detailed clarifications. If you have any remaining questions, please feel free to seek clarification. If our responses address your concerns adequately, we would sincerely appreciate your consideration in revising the assessment scores for our submission.
>
> Thank you, the Authors

---

### Official Review · Reviewer_fsJ1 · 2024-11-03

**Soundness:** 3
**Presentation:** 2
**Contribution:** 2
**Rating:** 6
**Confidence:** 3

**Summary:**

This paper introduces a new task called Mathematical Program Modification (MPM), where a mathematical program is revised in response to a natural language inquiry, referred to as "what-if" questions. The authors also examine how problem complexity, specifically the number of variables, impacts model performance. Experiments with the Llama-3.1 series show that more challenging problems remain difficult for large language models, indicating significant room for improvement in handling complex modifications.

**Strengths:**

- The task is novel, challenging, and highly relevant for real-world applications.
- The dataset is thoughtfully designed, with components like problem specifications that enhance clarity and usability.
- The task is easily scalable; complexity can be adjusted by increasing the number of variables, as demonstrated in this work.
- The task can be effectively generated and tackled by large language models like LLaMA, achieving promising accuracy levels.

**Weaknesses:**

My main concern lies in the evaluation aspect:
- The paper evaluates only three models from the LLaMA series. As a benchmark, this is somewhat limited. Could the authors include evaluations from more advanced models, such as GPT-4 or Claude-3.5? This would give the community a clearer understanding of model performance on this task.
- The what-if questions are generated by LLaMA models and then evaluated by similar models, which may introduce potential biases. Including results from a broader range of models would help clarify whether such biases are present.
- The paper could benefit from additional analyses, such as self-consistency performance and pass@k accuracy. These insights could add further depth to the evaluation.

**Questions:**

See weakness.
And the right part in figure1. is vague.

---

> ### Author Response · Authors · 2024-12-01
> **Response to Reviewer fsJ1 Comments**
>
> Thank you for taking the time to review our work and for highlighting its key strengths. We are pleased that the novelty, challenge, and real-world relevance of the task have been recognized. We also appreciate your acknowledgment of the thoughtfully designed dataset, which includes components like problem specifications to enhance clarity and usability. Additionally, we’re glad the scalability of the task—allowing complexity to be adjusted by increasing the number of variables—and its potential for effective application with large language models like LLaMA have been appreciated. These insights affirm the importance and practicality of our contributions, and we welcome the opportunity to address your feedback in more detail. Please find below some clarifications to your concerns:
>
> ### Weakness 1
>
> Thank you for this suggestion. We agree that including evaluations from more advanced models like GPT-4 or Claude-3.5 could provide valuable insights and further strengthen the benchmarking of our task. To address this concern, we have expanded our evaluations to include GPT-4. GPT-4 was tested using both random few-shot exemplar (without CoT) selection and similarity-based CoT settings. The results showed consistent trends with those observed for the Llama-3.1 models, though GPT-4 demonstrated superior overall performance. These new findings, along with detailed performance statistics, have been added to Table 3 of the revised paper and are also given below.
>
> | Model   | Setting                        | Accuracy (%) |
> |---------|--------------------------------|--------------|
> | GPT-4   | Zero shot                     | 63.15        |
> | GPT-4   | Few shot                      | 72.00        |
> | GPT-4   | Few shot CoT                  | 76.76        |
> | GPT-4   | Random few shot - 5 augmented variables | 44.44        |
> | GPT-4   | Similarity few shot CoT - 5 augmented variables | 54.83        |
>
> ### Weakness 2
>
> We understand the concern for bias from generation and evaluation from the same model. Although we have now included results with GPT-4, we would like to highlight that the what-if questions in our task are very simple, short, and objective in nature. Specifically, they only contain the required change and the target value or modification, leaving minimal room for bias induction.
>
> Importantly, these what-if questions are dictated by the heuristic modifications we make algorithmically to the original formulation matrices. This approach gives us significant agency in ensuring the clarity and objectivity of the questions, as they are directly tied to precise, deterministic changes in the mathematical formulations. For example, a typical what-if question might be:
> "What if the capacity constraint is increased to 500 units?"
>
> We hope that the results with GPT-4 further cement this argument.
>
> ### Weakness 3
>
> Thank you for the suggestion regarding pass@k accuracy and self-consistency performance.
> | Model                 | Setting                                  | Accuracy (%) | p@5 Accuracy (%) |
> |-----------------------|------------------------------------------|--------------|------------------|
> | Llama-3.1-70B Instruct | Few-shot (Random), on N=100 test set samples | 55          | 66              |
> | Llama-3.1-70B Instruct | Few-shot (Random), on N=396 test set samples | 64.64       | 71.96           |
>
> However we can see that the increase from exact-match accuracy to p@5 is not that significant, which may indicate that the relevance of such metrics in the context of the MPM task is limited.
> Metrics like pass@k are particularly suited to tasks such as code generation, where there are often multiple valid solutions, and diversity in responses is desirable. In such cases, evaluating whether at least one of the generated outputs is correct provides meaningful insights. However, MPM tasks are highly deterministic, with only one correct modification expected based on the specified what-if question. Thus, while pass@k provides some insight into how models explore the solution space, it is not as directly aligned with the task objectives as exact match accuracy, which measures whether the model produces the precise required modification.
>
> We plan to extend our evaluations to larger datasets in the future to further assess these metrics' applicability, but we hope this additonal evaluation data point addresses the reviewer’s concerns and provides clarity on the model’s performance nuances.

---

> ### Author Response · Authors · 2024-12-02
> **Response to Reviewer fsJ1 Comments**
>
> ### Questions
>
> We appreciate the reviewer's comment regarding the right part of Figure 1 and would like to clarify its contents.
>
> The right portion of the figure represents the **modified mathematical program P***, which results from applying the what-if question to the original mathematical program P. Specifically:
>
> - The row highlighted in red in the constraints matrix has been updated. The upper bound for the truck trips is modified from 5 to 10, as specified by the what-if question: *"What if the truck can make 10 trips per day?"*
> - This update reflects the adjustment in the mathematical program, showing how the original constraint on truck trips changes to accommodate the new upper bound.
> - The ellipsis (\(...\)) in the matrix indicates potential additional variables or constraints not shown explicitly for simplicity, as they remain unchanged by the what-if question.
>
> We hope this explanation resolves the concerns about the figure's clarity, we will try to make it more clear in a future version. Thank you for pointing this out.

---

> > ### Comment · Reviewer_fsJ1 · 2024-12-02
> > **Response to author**
> >
> > Thanks for your responses, which addressed most of my concerns. I have decided to raise my score. Thanks!

---

> > > ### Author Response · Authors · 2024-12-02
> > > **Official Comment by Authors**
> > >
> > > Dear reviewer,
> > > Thank you for updating your score—we’re glad we could address your concerns! If there’s anything else we can clarify or discuss to help you reconsider your score further, please let us know.

---

### Author Response · Authors · 2024-12-04
**A summary of the discussion phase for the perusal of the PCs and ACs**

## Strengths
All reviewers agreed that the work presents a novel and valuable contribution to the field, with particular praise for the WIQOR dataset. They highlighted the dataset’s relevance for evaluating large language models (LLMs) in the context of mathematical program modification (MPM), noting its potential to address real-world challenges in optimization. Reviewers appreciated the thoughtful design of the dataset, including its scalability and the use of human evaluation and heuristics to ensure data quality. They also acknowledged that the evaluations underscored the difficulty of the task and demonstrated the potential of LLMs in handling complex optimization problems. Overall, the work was recognized as a significant and original contribution to the field, offering a challenging task for current LLMs with broad implications for optimization and natural language processing.

## Weaknesses, questions and their Rebuttals
### Reviewer fsJ1:

- Reviewer pointed out the limited evaluation with only three LLaMA models, so we we expanded our experiments to include GPT-4 in zero-shot, few-shot, and CoT settings. The results showed that GPT-4 demonstrated superior performance compared to LLaMA models, supporting our conclusions.

- Reviewer raised concerns about potential bias from using the same model for generating and evaluating the "what-if" questions. We clarified that the questions are deterministic and based on algorithmic modifications, minimizing bias. We validated this by including GPT-4 evaluations, which showed similar trends, confirming the fairness of the evaluation.

- Reviewer suggested including additional evaluation metrics like pass@k accuracy and self-consistency performance. We tested Llama 3.1 70B Instruct on the task with pass@5 and presented the results although further clarified that exact-match accuracy is more suitable for this deterministic task, as it directly measures the correctness of the formulation modifications. Pass@k is more appropriate for tasks with multiple valid solutions.

- Reviewer asked for clarification on Figure 1. We clarified that the right side of Figure 1 shows the modified constraint, with the ellipsis indicating unchanged variables.

**Outcome**: Reviewer was satisfied with the rebuttal and increased their score to 6.

---

### Reviewer 6sCn:

- Reviewer mentioned the limited novelty in the data generation method and mode collapse concerns. We clarified that novelty of the data generation method was not the goal of our work, the method uses heuristics and human evaluation to ensure high-quality data. Mode collapse was assessed using the Self-BLEU score (0.50156), showing enough diversity in the generated questions.

- Reviewer expressed concerns about the exact-match accuracy being too restrictive and suggested solver-based evaluation. We clarified that exact-match accuracy directly measures the correctness of modifications, making it the most appropriate metric for this task. Solver-based evaluation is more suited for solution feasibility, which is not the focus here.

- Reviewer asked about the lack of extension to mixed-integer programming (MIP). We clarified that our current work focuses on linear programming, but our methods can be adapted for MIP in future research.

- Reviewer questioned the realism of the dataset. We clarified that the WIQOR dataset, based on NL4Opt, tests reasoning in practical optimization tasks. While not real-world problems, they are designed to assess model performance in structured problem-solving tasks.

**Outcome**: Reviewer didn't engage further and did not change their score.

---

### Reviewer mpi3:

- Reviewer pointed out draft-like presentation and verbosity. We clarified that we improved the paper’s formatting (e.g., Figure 6) and made Sections 2 and 3.1–3.2 more concise, focusing on key concepts while removing unnecessary details.

- Reviewer expressed concerns about the limited evaluations and lack of zero-shot and CoT ablations. We clarified that we included GPT-4 evaluations, including zero-shot results. The paper already had an ablation study for LLaMA-3.1, strengthening our conclusions.

- Reviewer asked for clarification on the motivation for using specifications. We clarified that specifications mirror real-world optimization workflows, where problems are modified iteratively using high-level summaries instead of re-writing the entire problem (Sections 3.1 and 4.2).

- Reviewer noted the absence of an NL-to-MP baseline comparison. We clarified that our approach of modifying existing formulations aligns with practical workflows (which use specifications instead of NL problems), ensuring continuity and efficiency, unlike regenerating the MP for each modification, which would be inefficient and inconsistent.

**Outcome**: Reviewer didn't engage further and did not change their score.

**Chairs are encouraged to consult the detailed rebuttals for a deeper understanding of the points above**

---

### Meta-Review · Area_Chair_NQmo · 2024-12-23

**Metareview:**

The paper introduces WIQOR, a novel dataset for the task of Mathematical Program Modification (MPM), where optimization problems are revised based on natural language "what-if" questions. The task is novel, challenging, and aligns with real-world workflows, and experiments demonstrate its potential to advance NLP and operations research. Weaknesses include limited evaluation on a narrow set of models (mostly LLaMA variants, with GPT-4 added later), as well as concerns about the validity and diversity of generated problems. The presentation and writing are not well-prepared and complained by some reviewers. This paper receives overall negative scores of 6,3,5. Unfortunately, several reviewers did not participate actively in the discussion, and I have read the reviews and the authors’ comments. I think the authors have done a good job in the rebuttal, where they clarified some concerns on “novelty” and some unreasonable complaints on the metric. My major concern is on the limited evaluation present – as a benchmark paper, I think the current evaluations are not enough even though the authors added GPT-4 during rebuttal, more models such as mistral and qwen are required to be more comprehensive. Also, the writing and presentation of the paper has a large room to be improved. Therefore, I would like to recommend rejection of this paper.

**Additional Comments On Reviewer Discussion:**

Reviewers praised the novelty and relevance of the WIQOR dataset but raised concerns about limited evaluations, potential bias in data generation, lack of solver-based evaluations, and the dataset's real-world applicability. Authors addressed these by expanding experiments to include GPT-4, demonstrating consistent trends and improved performance. They clarified the deterministic and objective nature of the what-if questions to minimize bias, justified the use of exact-match accuracy over solver-based metrics for this task, and provided evidence of data diversity using Self-BLEU scores. The authors addressed most of the concerns, but only several llama variants and GPT-4 are evaluated in this paper, which I think is not sufficient in a benchmark paper. The writing and presentation also need to be improved.

---

### Decision · Program_Chairs · 2025-01-22

Reject